# Artificial spider silk from ion-doped and twisted core-sheath hydrogel fibres

Yuanyuan Dou[1], Zhen-Pei Wang [2], Wenqian He[1], Tianjiao Jia[1], Zhuangjian Liu [2], Pingchuan Sun [1], Kai Wen[1,3], Enlai Gao [4], Xiang Zhou[3], Xiaoyu Hu[1], Jingjing Li[1], Shaoli Fang[5], Dong Qian[6] & Zunfeng Liu [1*]

Spider silks show unique combinations of strength, toughness, extensibility, and energy absorption. To date, it has been difficult to obtain spider silk-like mechanical properties using non-protein approaches. Here, we report on an artificial spider silk produced by the water-evaporation-induced self-assembly of hydrogel fibre made from polyacrylic acid and silica nanoparticles. The artificial spider silk consists of hierarchical core-sheath structured hydrogel fibres, which are reinforced by ion doping and twist insertion. The fibre exhibits a tensile strength of 895 MPa and a stretchability of 44.3%, achieving mechanical properties comparable to spider silk. The material also presents a high toughness of 370 MJ m$^{-3}$ and a damping capacity of 95%. The hydrogel fibre shows only ~1/9 of the impact force of cotton yarn with negligible rebound when used for impact reduction applications. This work opens an avenue towards the fabrication of artificial spider silk with applications in kinetic energy buffering and shock-absorbing.

[1] State Key Laboratory of Medicinal Chemical Biology, College of Pharmacy, Key Laboratory of Functional Polymer Materials, Nankai University, 300071 Tianjin, China. [2] Institute of High Performance Computing, A*STAR Research Entities, Singapore 138632, Singapore. [3] Department of Science, China Pharmaceutical University, 211198 Nanjing, Jiangsu, China. [4] Department of Engineering Mechanics, School of Civil Engineering, Wuhan University, 430072 Wuhan, Hubei, China. [5] Alan G. MacDiarmid NanoTech Institute, University of Texas at Dallas, Richardson, TX 75080, USA. [6] Department of Mechanical Engineering, University of Texas at Dallas, Richardson, TX 75080, USA. *email: liuzunfeng@nankai.edu.cn

Natural structural materials such as biofibres and biocomposites have shown extraordinary mechanical performance through the process of evolutionary selection[1]. Super-strong and tough structures that mimic of natural structural materials are highly sought-after, because of their desirable strength and texture properties[2]. Spider silk, a typical high-performance natural fibre, displays a specific combination of properties, i.e., high strength, large extension, and high damping capacity, resulting in higher toughness compared to other fibre materials[3–10]. Mimicking the structural characteristics of spider silk will allow new designs of novel fibre materials that can be used for energy absorption and impact reduction[11].

Tremendous efforts have been deployed to understand the structure of spider silk and to reproduce its mechanical properties using artificial fibres[12]. Biochemical and chemical methods have led to the development of artificial spider silks, such as protein fibres[13], supramolecular hydrogel fibres[14], and carbon nanotube (CNT) composite fibres[15]. These approaches have been proven to be successful to reproduce or partially reproduce the mechanical properties of spider silks. The CNT composite fibre made of CNT and silk protein shows a breaking strength of 600 MPa, a breaking strain of 73%, and a toughness of 290 MJ m$^{-3}$[16]. The supramolecular hydrogel fibre shows a breaking strength 193 MPa, a breaking strain of 18%, and a toughness of 22.8 MJ m$^{-3}$[14]. The artificial silk fibres based on regenerated silk proteins are the most widely studied, and promising mechanical properties (breaking strength ~1.34 GPa, breaking strain of ~36%, and toughness of ~334 MJ m$^{-3}$) have been obtained (Supplementary Table 1).

Although these fibres based on regenerated silk proteins approach the mechanical properties of natural spider silks (breaking strength of ~1.6 GPa, breaking strain of ~80%, and toughness of ~350 MJ m$^{-3}$), it is still difficult to prepare artificial spider silks using a non-protein approach. This may result from a poor understanding of the key structural characteristics of natural spider silks that are responsible for their mechanical properties and the difficulties faced when combining different structural models, such as amorphous regions crosslinked with crystallites[6], spiral nanofibres[7], and skin-core structures[8], to prepare artificial spider silks.

Spider silk is a natural hydrogel fibre that may strongly depend on water to form[11]; however, water is rarely considered when fabricating synthetic artificial spider silk. Previous studies have shown that spider silk exhibits a skin-core structure[17]. The core, which comprises an elastic inner centre surrounded by a plastic outer layer, is the key part to provide the mechanical properties of the silk, with the skin layer affording some protection against environmental impact[18].

Building on this inner- and outer-core architecture of natural spider silk, we fabricate artificial spider silk fibres from a hierarchical core-sheath structure obtained by the water-evaporation-controlled self-assembly of a polyacrylic hydrogel. The fibre contains hydrogen bonding and covalent networks, with the key mechanism being difference between the water-evaporation rates of the fibre core and the fibre sheath. The hydrogel consists of polyacrylic acid crosslinked with vinyl-functionalised silica nanoparticles (VSNPs). The fibres are made using a drawing method from the above hydrogel using a steel rod, and they are reinforced by adding ions into the hydrogel (called ion doping) and inserting twist into the fibres. The fibre exhibits a tensile strength of 895 MPa, a stretchability of 44.3%, a modulus of 28.7 GPa, a high toughness of 370 MJ m$^{-3}$, and damping capacity of 95%, which display comparable mechanical properties to natural spider silk.

## Results
### Fabrication and core-sheath structure of hydrogel fibres.
Briefly, the hydrogel was synthesized as follows (Fig. 1).[19,20] First,

VSNPs were prepared by adding vinyl-triethoxysilane (20 mM) to deionised water (30 mL), which was stirred for 12 h at room temperature. Acrylic acid (0.16 M) and ammonium persulfate (0.1 mM) were added to the dilution of the VSNP dispersion (18 mL), which was then stirred for 30 min for complete mixing. Then, the mixture was allowed to react for 30 h under $N_2$ protection at 40 °C to obtain the hydrogel. The fibres can be formed with a wide variety of feed ratios of VSNPs, i.e., from 0.1 wt to 0.5 wt% (Supplementary Table 2), and the fibre diameter can be tuned from 10 to 500 μm, which was linearly proportional to the dipping depth of the drawing rod (Supplementary Fig. 5a). The as-drawn wet fibre was significantly viscoelastic, and a thin fibre (e.g., 20-μm in diameter) will return to approximately half its length if the stretch was immediately released. Exposing the wet fibres to ambient air for a certain time (10–500 s), i.e., by keeping both ends tethered to the drawing rod and the bulk at a constant length, fixed their shape by water evaporation, and the fibres remained stable without any change in the fibre length. The resulting hydrogel fibres showed a transparent core and an opaque sheath under confocal microscopy (Fig. 1d, e) and under metallographic microscopy in a reflective mode (Supplementary Fig. 5c). The sheath appeared opaque by optical microscopy because light scattering increased at a reduced water content. With increasing the drying time, the sheath thickness increased while the core diameter decreased (Fig. 1d, Supplementary Fig. 5c), indicating lower water content in the sheath than in the core. This increased hydrogen bonding between the polymer chains and decreased the competitive binding of the water molecules, as confirmed by Fourier transform infrared spectroscopy (FTIR) (Supplementary Fig. 6a) and nuclear magnetic resonance (NMR) (Supplementary Fig. 6b), which resulted in the higher elastic modulus of the fibre sheath than the fibre core. When the non-elastic sheath and the stretched elastic core reached a mechanical balance, the hydrogel fibre length remained unchanged upon tethering removal. Therefore, hydrogel fibres with larger diameters required a longer setting time (minimum drying time required for the fibre to reach a mechanical balance) than their thinner counterparts after fibre drawing (Supplementary Fig. 7).

As the hydrogel fibres are exposed to environments in ambient air (40% humidity), and the sheath is formed from the fibre surface due to water molecule evaporation. From the material point of view, the sheath and the core are formed by the same polymer, with less water content in the sheath compared to the core. Therefore, this core-sheath structure is formed spontaneously when the fibre dries in ambient air. To check if this core-sheath structure also presents in other hydrogel materials or if this drawing technique is necessary for this core-sheath structure, we prepared a polyacrylamide/alginate hydrogel according to the literature[21], and prepared a 375-μm-diameter fibre using a polypropylene tube as the template. Core-sheath structures were also observed under metallographic microscopy in the reflective mode (Supplementary Fig. 18a). This indicates that the core-sheath structure of the hydrogel fibre can also be observed in other hydrogel materials and without using a drawing technique.

To further understand the effect of water on the core-sheath structure of the hydrogel fibre, we characterised the surface morphology changes under exposure to water moisture. Similar to the supercontraction behaviour of spider silk[22], the hydrogel fibre contracted when exposed to water moisture. The supercontracted hydrogel fibre showed a buckled surface morphology (Supplementary Fig. 8a–c). This likely results from the crumpling of the plastic sheath over the elastic core during the release of the pre-strain of the fibre core, which stems from the moisture-induced decrease in the sheath modulus. The amount of water vapour (20% fibre weight) that was enough to initiate supercontraction by

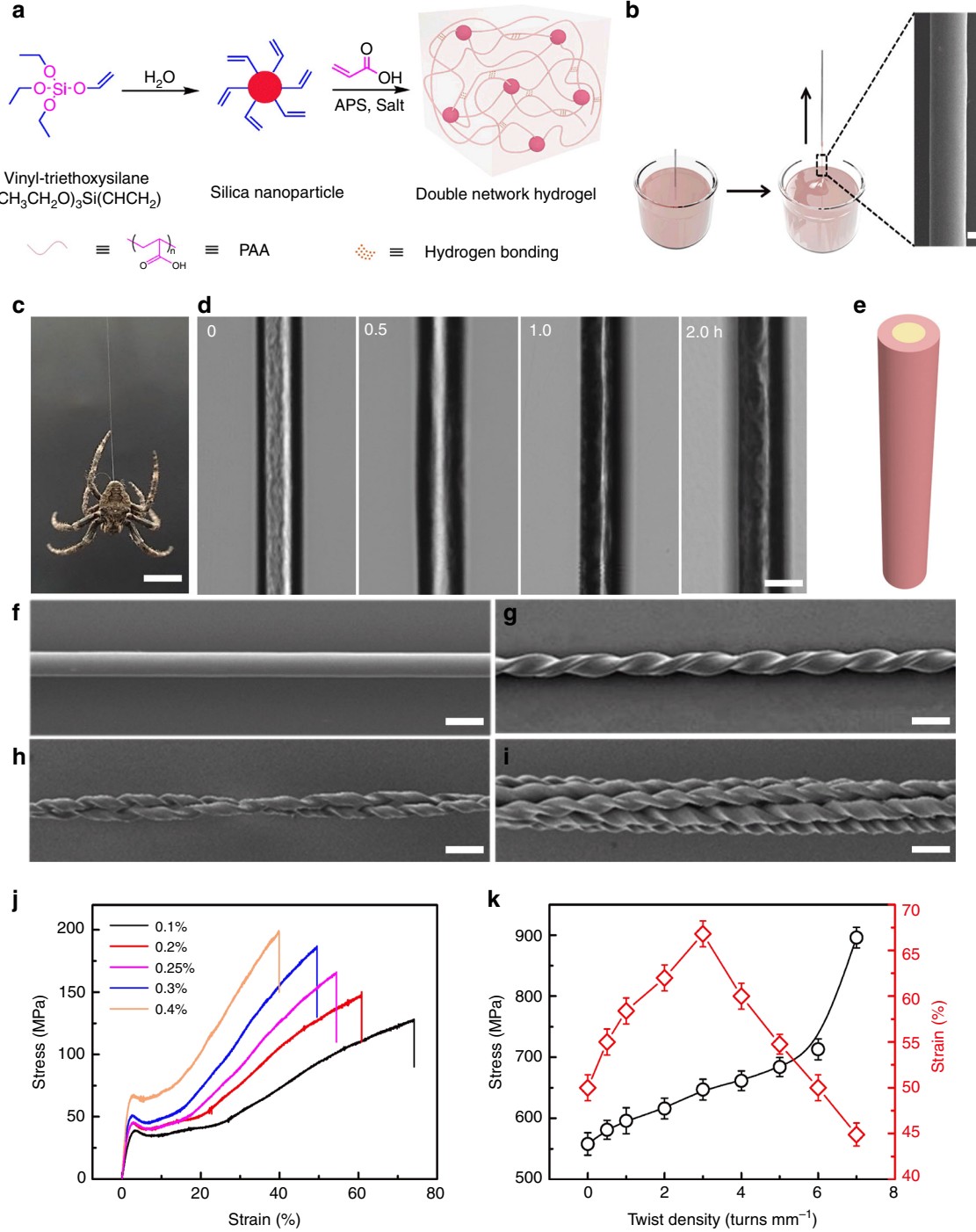

**Fig. 1** Preparation, morphology and mechanical strength of hydrogel fibres. **a** Two-step synthesis of hydrogel involving VSNP formation and free-radical polymerisation. **b** A single fibre was drawn by dipping a steel rod into a hydrogel reservoir. The diameter of the steel rod was 0.2 mm, and the drawing speed was 4 cm s$^{-1}$. Scale bar: 10 μm. The hydrogel fibre shown in the micrograph contained 0.1 wt% VSNPs, without salt or inserted twist, and had a diameter of 20 μm. It was obtained at an atmospheric relative humidity of 40%. **c** An *Araneus diadematus* spider hanging on a strand of natural spider silk; the scale bar is 2 cm. **d** Laser confocal microscope images of hydrogel fibres with different drying times. Scale bar: 20 μm. **e** Core-sheath model of the hydrogel fibre. **f–i** SEM images of non-twisted (**f**), twisted (**g**), self-balanced, 2-ply (**h**), and self-balanced, 6-ply hydrogel fibres (**i**). The scale bars for **d–i** are 20 μm. The twist density was 7 turns mm$^{-1}$. **j** Tensile stress-strain curves of the as-drawn hydrogel fibres with different VSNP contents after shape setting. The hydrogel fibres contained 0.1–0.4 wt% VSNPs, without inserted twist or salt. The deformation rate was 1.1% s$^{-1}$. **k** Breaking stress and breaking strain of hydrogel fibres with optimised mechanical properties as a function of twist density at a deformation rate of 27.8% s$^{-1}$. In **d**, **f–i** and **k**, the hydrogel fibres contained 0.1 wt% VSNPs and 20 mM ZnCl$_2$. The error bars mean the s.d. from five measurements

decreasing the fibre-sheath modulus did not concomitantly cause volume expansion (Supplementary Fig. 8e). Interestingly, the buckled surface of the supercontracted hydrogel fibre became smooth when exposed to humidity for over three minutes (Supplementary Fig. 8d), indicating that the fibre sheath became elastic and released the internal stress.

**Mechanical strengthening of hydrogel fibres.** The core of the natural spider silk, which contains spiral nanofibrils of 30–35 nm in diameter, provides the mechanical strength[7]. Each nanofibril is composed of amorphous peptide regions crosslinked with nano β-crystallites[23]. It has been proposed that the combination of these structures plays a key role in producing the high mechanical properties of spider silk[23,24]. Inspired by this, we reinforced the core-sheath hydrogel fibres by metal doping to increase the crosslinking sites between the polymer chains and by twist insertion to provide a spiral architecture of fibres. If not specified, a 20-μm-diameter hydrogel fibre was used for the following investigations. The hydrogel fibre that contained 0.1 wt% VSNPs showed a tensile strength of 127 MPa and a breaking strain of 73.1%; increasing therosslinking density to 0.4 wt% enhanced the tensile strength to 197 MPa and reduced the breaking strain to 40% (Fig. 1j). By adding metal ions ($Zn^{2+}$, $Mg^{2+}$, $Na^{+}$, $K^{+}$) into the hydrogel fibres, a significant increase in tensile strength was observed. Adding 20 mM $Zn^{2+}$ to the hydrogel fibre significantly increased the tensile strength to 261 MPa (Supplementary Fig. 9a, b). The FTIR measurements showed that $Zn^{2+}$ increased the crosslinking between the polymer chains (Supplementary Fig. 9c, d). These results are consistent with previous findings, suggesting that doping with metal ions can improve the mechanical properties of hydrogels by increasing the crosslinking between polymer chains[21]. Elemental mapping shows that $Zn^{2+}$ ions were distributed evenly in the cross-section of the hydrogel fibre (Supplementary Fig. 5b). Increasing the deformation rate (stretching speed) of the $Zn^{2+}$-doped fibre from 1.1% $s^{-1}$ to 27.8% $s^{-1}$ improved the tensile strength from 261 to 514 MPa with a negligible decrease in the breaking strain (Supplementary Table 3).

We measured the $^1H$-$^1H$ double-quantum (DQ) NMR spectra of the samples to probe the hydrogen bonding structure of PAA when the metal ion was included in the system. First, the $^1H$ magic-angle spinning (MAS) spectra of the samples with and without $ZnCl_2$ under 40 kHz fast MAS were measured, as shown in Supplementary Fig. 9e. Proton signals of different types of COOH dimers, free COOH and PAA were observed for the sample without $ZnCl_2$. With the addition of $ZnCl_2$, the peak of the COOH dimer obviously decreased, and a new strong peak at approximately 7.9 ppm was observed, this new peak can be assigned to the protons undergoing a chemical exchange between free COOH and water, as reported in our previous work[25].

Next, we measured the $^1H$ DQ NMR spectra of the samples without and with $ZnCl_2$ under 40 kHz MAS, as shown in Fig. 2e, f. In Fig. 2e, for the sample without $ZnCl_2$, a strong diagonal peak at 13 ppm was observed, corresponding to the formation of hydrogen-bonded COOH dimers, and cross peaks between the protons of the COOH dimer and the aliphatic protons of PAA were clearly observed. In Fig. 2f, with the addition of $ZnCl_2$, the diagonal peak at 13 ppm of COOH dimers and the cross peaks between the COOH dimer and aliphatic protons (thick red line in Fig. 2f) obviously decreased, indicating the destruction of the hydrogen-bonded COOH dimers. With the application of a DQ filter in the $^1H$ DQ experiment, the strong proton signals under chemical exchange at 7.9 ppm shown in Supplementary Fig. 9e were greatly reduced due to their relatively high mobility, while the less mobile free COOH signal was retained. Predominant

cross peaks between the protons of free COOH and aliphatic groups of PAA were observed (thick red line in Fig. 2f). In short, the 2D $^1H$ DQ NMR spectra clearly prove the destruction of the hydrogen bonding of PAA when the metal ion was included in the system.

Spiral nanofibril structures have been observed in spider silk and may be important for the silk's mechanical properties[24]. Introducing a helix in the hydrogel fibre by twist insertion at 7 turns $mm^{-1}$ (Fig. 1g) boosted the tensile strength to 895 MPa (Fig. 1k) and the modulus to 28.7 GPa, which are almost 1.7 and 1.9 times, respectively, those for the non-twisted fibre (514 MPa for strength and 14.7 GPa for modulus), and the stretchability slightly decreased to 44.3%, compared to the non-twisted fibres (64.7%) (Fig. 2b). Releasing the tethering of a twisted, non-strained fibre did not result in untwisting of the fibre; however, it did partially untwist under a load. Folding a twisted fibre onto itself by bending it at mid-length resulted in the fibres being plied together upon untwisting of each fibre and a self-balanced fibre was produced (Fig. 1h and Supplementary Table 3). The mechanical strength of the self-balanced fibre increased from 640 to 840 MPa and the stretchability decreased from 47.1 to 33.2% when the number of plies was increased from two to eight (Fig. 1i and Supplementary Fig. 10a). Increasing the number of plies from 10 to 100 decreased both the breaking stress and breaking strain (Supplementary Table 4, and Supplementary Fig. 10e). These twisted hydrogel fibres exhibited mechanical strength and stretchability values that exceeded those of conventional regenerated textile fibres, such as cellulose-based viscose (350 MPa, 4%)[26–28], as well as animal and human hairs (320 MPa, 66%)[29]; these hydrogel fibres also had similar strength and flexibility compared to the silk-protein-based fibres (1.34 GPa, 36%)[12] and silk protein/CNT composite fibres (600 MPa, 73%)[16].

Environmental humidity affects the mechanical properties of hydrogel fibres. Figure 2a shows that when the humidity increased from 40 to 100% at room temperature, the tensile strength decreased from 895 to 180 MPa and the strain at break increased from 44.3 to 143%. The mechanical properties of the hydrogel fibres also greatly depend on the fibre diameter, as shown in Supplementary Fig. 10b. When decreasing the fibre diameter from 300 to 20 μm, the maximum available strain decreased from 200 to 40% (80% decrease), and the strength increased from 2.8 to 237 MPa (83-fold increase). This phenomenon is mainly caused by the following: (i) the water evaporation potential increases with an increase in the surface curvature (Kelvin curvature effect), causing the stiffness of fibres with smaller diameters to mainly be dominated by the fibre sheath; (ii) hydrogel chains have the capability to absorb and store water, and the fibres with larger diameters have larger water storage capabilities, which restrains the sheath growth; (iii) there is an interface layer between the core and the sheath. Correspondingly, the ratio between the sheath thickness and the core diameter observed from images decreased from 62.5 to 40%, with the fibre diameter increasing from 20 to 60 μm (Supplementary Fig. 10c, d). The mechanical strength of a 50-μm-diameter hydrogel fibre rose monotonically from 9.8 MPa to 78.7 MPa when increasing the water evaporation time from 10 s to 7200 s (Supplementary Fig. 11a, b). With an increase in the drying time, we also observed an increased ratio of the sheath thickness to the core diameter (25–90%) (Fig. 1d). Similarly, as the drying time of a 375-μm-diameter polyacrylamide/alginate hydrogel fibre increased from 10 s to 7200 s, the mechanical strength also showed a dramatic increase from 0.439 to 1.08 MPa (Supplementary Fig. 18b), accompanied by an increased fibre-sheath thickness. These observations agree with the assumption that the core-sheath structured hydrogel fibre consists of a high-modulus plastic fibre-sheath and a low-modulus elastic fibre-core, which is similar to the structure of spider silk[8].

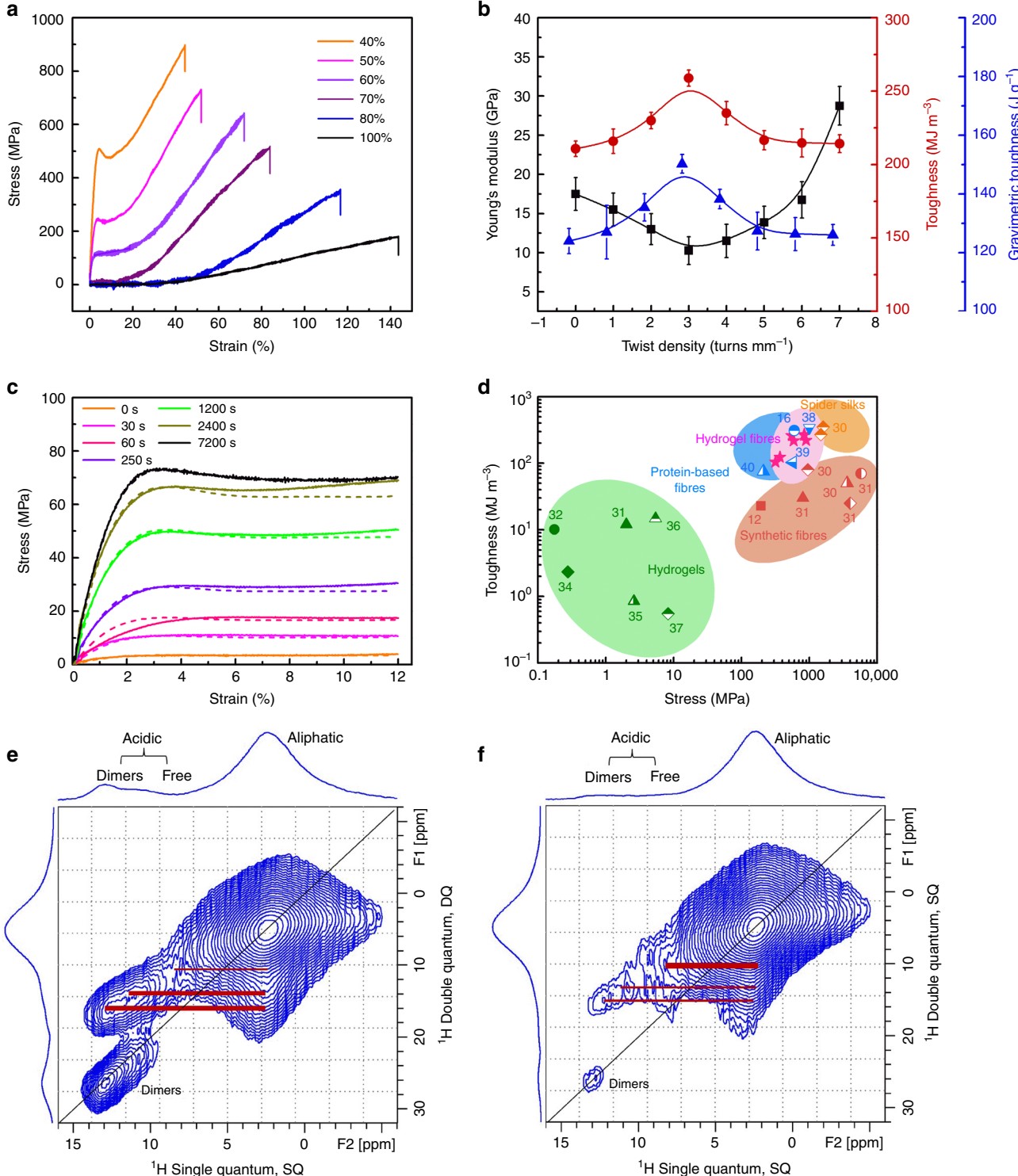

**Fig. 2** Modelling of the mechanical properties and toughness of the hydrogels. **a** Tensile stress-strain curves of hydrogel fibres with optimised mechanical properties under different humidity values. The twist density was 7 turns mm$^{-1}$, and the deformation rate was 27.8% s$^{-1}$. **b** The Young's modulus, toughness, and gravimetric toughness of the hydrogel fibres as a function of the twist density. The deformation rate was 27.8% s$^{-1}$. **c** FEM simulation results of the engineering stress-strain curves of the hydrogel fibres (dotted line) with the same diameter (50 μm) but different drying times; the solid curves represent the experimental data. **d** Comparison of the energy dissipation and damping capacity of hydrogel fibres (pink stars) in this work with those of other typical energy-dissipation materials, such as hydrogel (green symbols), spider silk (yellow symbols), protein-based fibres (blue symbols) and synthetic fibres (red symbols), reported in the literature. The numbers shown in the graphs correspond to the references. The hydrogel fibres presented different twist densities and were tested at different deformation rates. All the hydrogel fibres contained 0.1 wt% VSNPs and 20 mM ZnCl$_2$. **e, f** The $^1$H double-quantum/single-quantum (DQ/SQ) chemical shift correlation spectra at 40 kHz MAS of samples: **e** without and **f** with ZnCl$_2$. Two rotor periods of Back-to-Back (BABA) recoupling were used for the excitation and reconversion of the DQ coherence. The error bars mean the s.d. from five measurements

**Core-sheath structure modelling**. From the material point of view, the sheath and the core are formed by the same polymer, with less water content in the sheath compared to the core. The sheath is identified as its mechanical properties become higher than the core (swelled hydrogel). It is reasonable that with an increase in the drying time, some of the core is converted to the sheath, due to the desorption of water molecules. So a numerical simulation was carried out for the mechanical properties of hydrogel fibre at 40% humidity for different exposure times based on a core-sheath model, using a freshly drawn hydrogel fibre with a drying time of zero as the lower bound limit (fibre core) and using a fibre dried at 40% humidity for 2 h as an upper bound limit (Supplementary Note 5). The hydrogel fibre dried at 40% humidity for 2 h shows almost the same mechanical properties as the fibre dried for 4 h until the yielding point (Supplementary Fig. 10f). The metallographic microscope in the reflective mode shows that the hydrogel fibre exposed to ambient air for 2 h and 4 h almost completely dried (~85% of fibre-sheath ratio) (Supplementary Fig. 5c), which indicates that the water absorption/desorption in the fibre sheath with the ambient air almost reached an equilibrium.

This proposed core-sheath model was validated using the finite element method (FEM). The combination of a soft core with a low Young's modulus and a stiff sheath with a high Young's modulus leads to fibres with the same diameter but with different drying times and different mechanical properties. This allowed us to obtain good agreement of the numerical simulation of the stress-strain curves (before the yielding point) with the experimental data for hydrogel fibres with different exposure times, as shown in Fig. 2c. The sheath thicknesses for fibres with different drying times were theoretically calculated, which increased with the increase in the drying time. This indicates that the fibre core partially converted to the fibre sheath as the drying time increased. The theoretically calculated fibre-sheath thicknesses at different drying times are similar to the optically measured sheath thicknesses, which indicates that the fibre core partially converted to the fibre sheath as the drying time increased.

Both the numerical and experimental observations show that as the drying time increases, the increasing speed of sheath thickness formation decreases, which indicates that the sheath also performs as an isolator to prevent moisture from evaporating out of the fibre core. Therefore, the modulus of the fibres with larger diameters were lower than those for smaller-diameter hydrogel fibres when they were exposed to the same humidity levels for the same amount of time. The ambient air (40% relative humidity) contains water moisture, so there should be an absorption/desorption balance between the hydrogel fibre sheath with the ambient air.

**Characterisation and modelling of hydrogel fibre by twist insertion**. We first investigated the morphology change of hydrogel fibres during twist insertion, as shown in Supplementary Fig. 4a. It can be seen that with twist insertion, the hydrogel fibres get flattened, and the inner core becomes less transparent. This indicates that the fibre becomes more sheath-like, possibly due to the decreased thickness in the flattened regions. This corresponds to the increased mechanical stretch and modules of the hydrogel fibre with an increase in the inserted twist. Twist insertion results in an increase in the mechanical strength and modules of the hydrogel fibres. An FEM simulation was carried out to validate the effect of twist insertion on the internal residual stress/strain of a core-sheath structured hydrogel fibre. A 20-μm-diameter hydrogel fibre (sheath thickness of 3 μm and core radius of 7 μm) was constructed in FEM. It is found that upon twist insertion (6 turns mm$^{-1}$), periodic internal stress was generated

along the fibre length in both the fibre sheath and fibre core, resulting in twisting of the hydrogel fibre and flattening in some regions (Supplementary Fig. 4b). This agrees with the experimental observation of twist-induced flattening of hydrogen fibres in Supplementary Fig. 4a.

Moreover, the lengths of the polymer chains in the fibre were actually elongated by forming a spiral structure during twist insertion. This may result in the following two consequences at the macromolecular level. (1) The random coiled polymer chains were stretched and became aligned to some extent in the twisting direction. Such an alignment of polymer chains would result in an increased anisotropy of the fibre, increasing the mechanical strength during stretching. (2) Twist-induced elongation of the polymer chain may generate internal stress and increase the bond angle in the polyacrylic acid chain, therefore inducing increased rigidity of the polymer chains. These twist-induced changes at the macromolecular level may contribute to the increase in the mechanical strength.

**Fracture toughness and energy dissipation of hydrogel fibres**. Breaking energy or toughness is the energy required to break a material and depends on the combination of mechanical strength and stretchability. The toughness of the hydrogel fibres (259 MJ m$^{-3}$) was maximised by optimising the material composition, twist density, and deformation rate (Supplementary Tables 3 and 4). This toughness is comparable to that of natural spider silks, which ranges from 160 to 350 MJ m$^{-3}$ according to the type of silk[12,30,31]. This unique toughness is superior to those of existing hydrogel materials[32–37] and synthetic fibres including carbon fibre (25 MJ m$^{-3}$)[12,30], polyethylene (30 MJ m$^{-3}$)[31], Kevlar 49 (50 MJ m$^{-3}$)[12,30,31] and nylon 6, 6 (80 MJ m$^{-3}$)[12,30] and is close to those of the silk-protein-based fibres (334 MJ m$^{-3}$)[38,39] and silk protein/CNT composite fibres (290 MJ m$^{-3}$)[16,40] (Fig. 2d).

Next, we assessed the energy dissipation and damping capacity of the hydrogel fibres (Fig. 3). The energy dissipation of fibres corresponds to the energy absorbed by the fibres from an incoming impact. The damping capacity, which is the ratio of energy dissipation to incoming energy, was evaluated as the percent difference between the loading and unloading energies (Supplementary Fig. 11c and Note 2). A single hydrogel fibre was subjected to progressive loading/unloading cycles at strains of 1.6–40%. The damping capacity amounted to 64% at 1.6% strain for the first cycle and increased to ~95% for the following cycles at strains ranging from 5 to 40% (Supplementary Fig. 11d). This behaviour is superior to that of spider silk, which has a damping capacity that drops from ~68% in the first cycle to as low as 37% in subsequent cycles[14]. Interestingly, the damping capacity for the hydrogel fibres remained at ~95% regardless of the material composition, inserted twist, and deformation rate (Supplementary Fig. 12). This indicates that high energy dissipation can be obtained by improving the mechanical strength and stretchability with optimised factors, such as the material composition, inserted twist, and deformation rate (Supplementary Tables 3 and 4). Indeed, the polyacrylic hydrogel in this work contains hydrogen-bonding and covalently crosslinked networks, which is similar to the hydrogel work by *Suo* et al.[21]. The covalent network is formed by crosslinking of the vinyl groups in acrylic acid and in vinyl-functionalised silica nanoparticles, which can be deformed under stress and return to the initial state after stress relaxation. During deformation, the hydrogen-bonding crosslinks unzip and dissipate energy; after the gel recovers the shape, the hydrogen-bonding rezips again. By adding $ZnCl_2$, we observed a dramatic increase in mechanical strength. As proved by our NMR analysis, $Zn^{2+}$ forms ionic cross-links with the $COO^-$ groups on the polyacrylic chains (Fig. 2e, f, and Supplementary Fig. 9e). This is

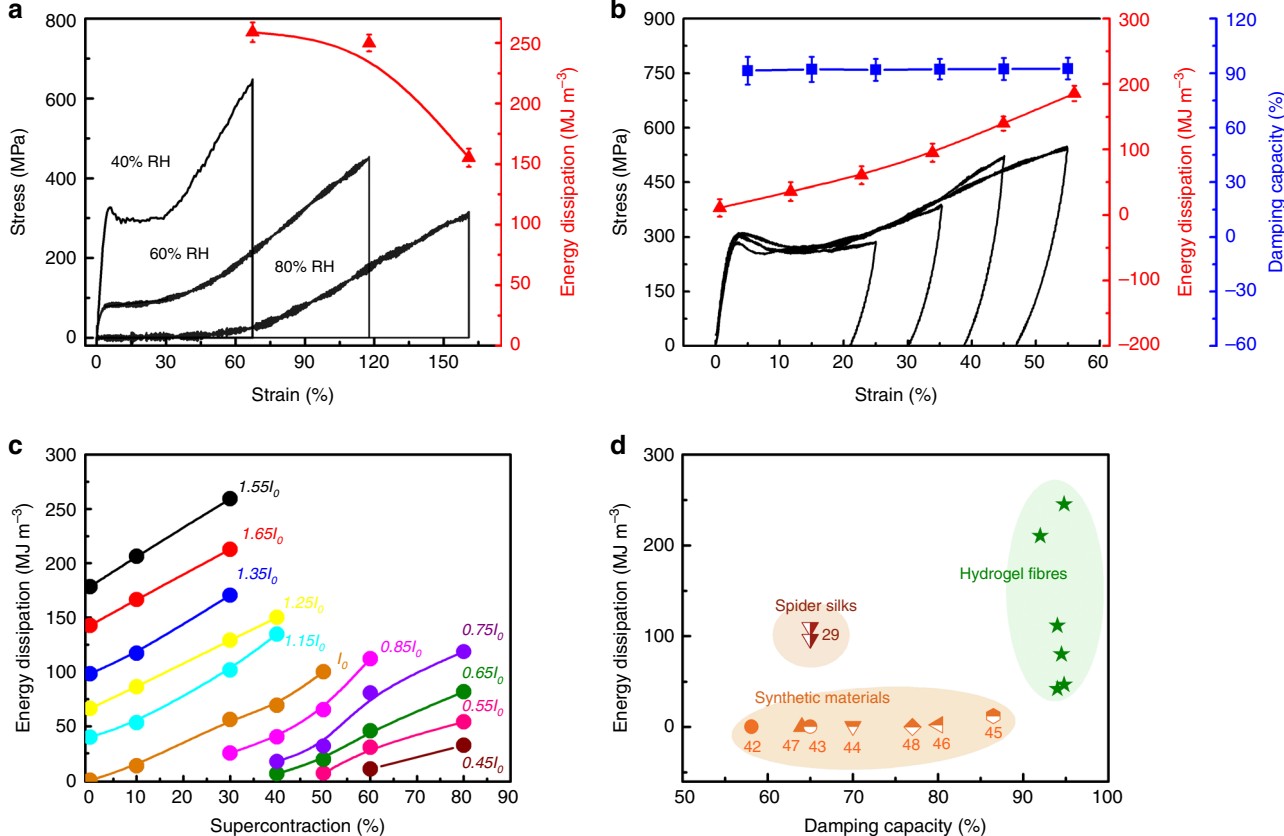

**Fig. 3** Energy-dissipation properties of hydrogel fibres. **a** Stress-strain curves and energy dissipation of hydrogel fibres at different humidity values. **b** Stress-strain curves, damping capacity, and energy dissipation of hydrogel fibres progressively stretched to different strains. **c** Energy dissipation of hydrogel fibres supercontracted to different degrees and stretched to different lengths. $l_0$: initial length of the hydrogel fibre. For **a–c** the twist density was 3 turns mm$^{-1}$, and the deformation rate was 27.8% s$^{-1}$. **d** Comparison of the energy dissipation and damping capacity of the hydrogel fibres (green stars) in this work with those of other typical energy-dissipation materials, such as spider silks (red symbols) and synthetic materials (yellow symbols), reported in the literature. The numbers shown in the graphs correspond to the references. The hydrogel fibres presented different twist densities and were tested at different deformation rates. All the hydrogel fibres contained 0.1 wt% VSNPs and 20 mM ZnCl$_2$. The studies with both the energy dissipation and damping capacity data reported are selected. The error bars mean the s.d. from five measurements

another ionic crosslink, which also dissipates energy during mechanical deformation by unzipping and rezipping after stress-relaxation.

Holding the stretched hydrogel fibre (at 40% RH) for three minutes resulted in negligible resilience, and an even higher damping capacity (~100%) and energy dissipation (~259 MJ m$^{-3}$) were obtained (Fig. 3a, Supplementary Table 4). When exposed to humidity levels exceeding 60% at room temperature, the as-drawn (20-μm-diameter) or extended hydrogel fibres could contract into a very small ball with diameter of 50 μm (Supplementary Fig. 13a–c), and the fibres can undergo eight contraction-stretching cycles (55% contraction) without loss of strength after five contraction-stretching pretreatment cycles (Supplementary Fig. 14a). The degree of super contraction was negligibly affected by the material composition, such as the VSNP content (Supplementary Fig. 13d). An energy dissipation of 260 MJ m$^{-3}$ was obtained by stretching a hydrogel fibre with a degree of supercontraction of 30% to 1.55 times its initial length (Fig. 3c). The hydrogel fibre maintained a high damping capacity and energy dissipation during progressive stretching-contraction cycles (Fig. 3b) and fully stretched-contraction cycles (Supplementary Fig. 14b). These results are superior to the damping capacity, are close to the energy dissipation of spider silk[41,42] and are higher than most existing synthetic energy-dissipation materials, such as hydrogels (~12.2 MJ m$^{-3}$, ~87%)[43–46], CNT foams (2.4 MJ m$^{-3}$, 80%)[47], CNT arrays (0.283 MJ m$^{-3}$, 64%)[48],

and metallic microlattices (3.5 KJ m$^{-3}$, 77%)[49] (Fig. 3d, Supplementary Figs. 15, 16 and Supplementary Tables 5 and 6).

Fibres capable of dissipating torsional energy and minimising torsional oscillation are highly desired for shock-absorbing cables. For example, a spider rarely rotates when hanging on natural spider silk[50]. To investigate the torsional energy damping properties, a hydrogel fibre was isobarically loaded (1.2 MPa) and torsionally tethered. Twist was inserted into the hydrogel fibre using a motor from the top, and then the load was allowed to rotate freely. When a 10-cm-long hydrogel fibre was twisted by 560° cm$^{-1}$ from its initial position, it oscillated slightly for three cycles around the new pseudo equilibrium position, and this oscillation behaviour did not show dependence on the twist density (Supplementary Fig. 17a). This dynamic torsional response of hydrogel fibres indicated a typical highly under-damped oscillation behaviour, which was similar to the torsional response of natural spider silk. In contrast, copper and nylon fibres with the same diameter, length, and twist density continued oscillating after ten damping cycles (Supplementary Fig. 17b).

**Impact reduction capacity of hydrogel yarn.** According to the current building safety standard for construction, a ledge positioned 1.8 metres or higher above a lower level should be protected with shock-absorbing threads or nets for impact reduction in case of accidental fall. Based on the unique combination of

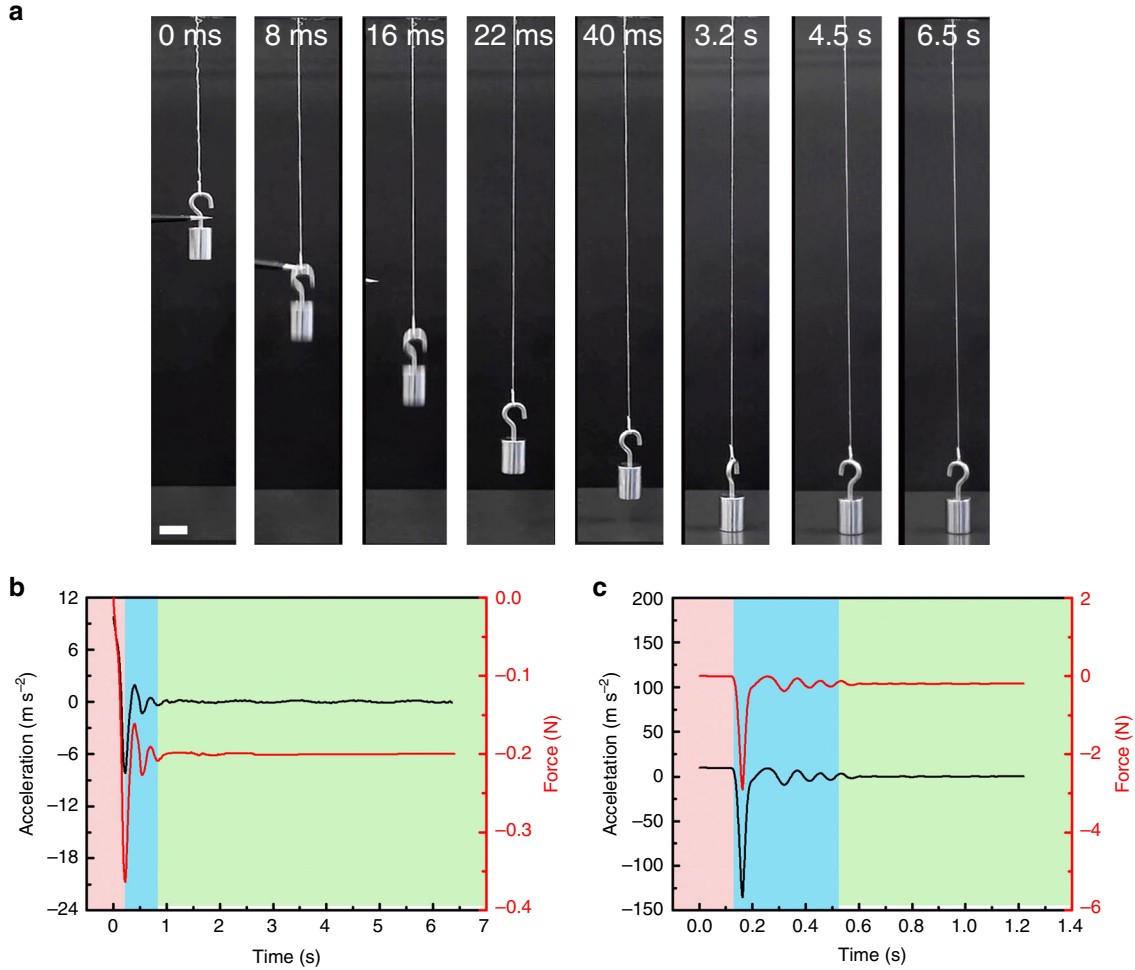

**Fig. 4** Impact reduction ability of hydrogel yarns. **a** Snapshots of a free falling 20-g object connected to a hydrogel yarn. The falling height was 15 cm. Scale bar: 2 cm. **b, c** Acceleration and impact force of the object buffered by (**b**) a hydrogel yarn and (**c**) a cotton yarn as a function of time. The hydrogel yarns were made of 100-plied, 10-cm-long, 20-μm-diameter hydrogel fibres and the hydrogel fibre contained 0.1 wt% VSNPs and 20 mM $ZnCl_2$ with an inserted twist of 3 turns $mm^{-1}$. The cotton yarn was 200 μm in diameter and 25 cm in length

high strength and stretchability with the high damping capacity of the hydrogel fibres, we investigated their capability to reduce the impact force and dissipate energy in shock-absorbing applications, such as vertically and horizontally placed shock-absorbing yarns as well as a net.

The hydrogel yarn must absorb the kinetic energy of a falling object without breaking and show low rebounding of the object back from the yarn. A 20-g weight (130 MPa in stress, without breaking the hydrogel yarn) was vertically tethered to a 10-cm-long yarn comprising 100-ply, 20-μm-diameter hydrogel fibres and was allowed to freely fall 15 cm at 60% RH (Fig. 4a and Supplementary Movie 1). The hydrogel yarn extended by 150% with negligible rebound and a maximum impact force of 0.36 N, which is 8.6 times less than that of a 15-cm-long cotton yarn used for an identical weight at the same height (Fig. 4b, c, Supplementary Fig. 1, Notes 3.1 and 3.2.1). In another configuration, the load was attached to a 30-cm-long polyethylene fibre and positioned in the middle of a horizontal hydrogel yarn before it was allowed to fall (Supplementary Movie 2). The weight bounced slightly for a few cycles before stopping, reaching a maximum impact force of 2.2 N, which is 9.1 times less than that of the cotton yarn (Supplementary Figs. 2, 3 and Note 3.2.2). In contrast, when the hydrogel yarn was replaced by a rubber fibre (high strength, high extensibility, and one of the most widely used synthetic material), the weight bounced up and down many times

before stopping (Supplementary Movie 2). The hydrogel yarn reached a toughness of 370 MJ $m^{-3}$ before breaking at a deformation rate of 12,000% $s^{-1}$ by extending the polyethylene fibre to eight metres (Supplementary Table 7). A shock-absorbing net knitted using the same hydrogel yarns underwent a 50% deformation and safely captured a 50-g egg freely falling from one metre at 60% RH. (Supplementary Movie 3). In contrast, an egg broke in the cotton net due to a high impact force, and an egg bounced out of the rubber net due to high rebound (Supplementary Movie 3). A deformed hydrogel net could retrieve its initial shape when exposed to high humidity (60% RH), indicating a reuse effect under water moisture (Supplementary Movie 4).

## Discussion

The combination of water-evaporation-induced self-assembly, ion doping, and twist insertion produces hierarchically structured hydrogel fibres. The fibres consist of a plastic sheath around an elastic core, leading to spider-silk-like strength and stretchability. Numerical analysis was used to help understand the core-sheath model. The twisted hydrogel fibres show highly under-damped oscillation behaviour under load; they also demonstrate outstanding toughness and impact reduction with low rebound. We believe that twisted core-sheath hydrogel fibres hold great

potential for energy dissipation and shock-absorbing applications, such as life-saving, high-rise escape ropes and nets, hanging ladders for helicopters, and parachute cords. This artificial spider silk material merits further study to realise its full potential.

## Methods

**Synthesis of hydrogel.** Vinyl-triethoxysilane (20 mM, Energy Chemical, Shanghai) was added to deionised water (1.6 M) under vigorous stirring for 12 h until a transparent dispersion of VSNPs was obtained. Then, acrylic acid (0.16 M, Alfa Aesar, Shanghai), ammonium persulfate (0.1 mM, Alfa Aesar, Shanghai), and metal chloride salts ($ZnCl_2$, $MgCl_2$, NaCl, or KCl, Aladdin, Shanghai, 10–50 mM) were added to the dilution of the VSNP dispersion (18 mL). The mixture was stirred for another 30 min at room temperature. The solution was degassed and sealed under $N_2$ to remove the dissolved oxygen and was allowed to undergo free-radical polymerisation for 30 h in a water bath at 40 °C to produce hydrogel.

**Fabrication of hydrogel fibre.** To generate the hydrogel fibre, a 0.2-mm-diameter stainless steel rod was vertically dipped into the reaction mixture and immediately drawn out at a speed of 4 cm s$^{-1}$. The fibre was solidified by holding both its ends and exposing the fibre to ambient air at 40% RH for 30–500 s. To introduce twists into an as-drawn hydrogel fibre, one of its ends was attached to an eighty-step servomotor, and the other end was torsionally tethered to an isobaric load of 1.2 MPa to avoid rotation.

## Data availability

All relevant data are available from the corresponding authors upon reasonable request.

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

## Acknowledgements
This work was supported by the National Key Research and Development Program of China (grant 2016YFA0200200, 2017YFB0307000), the National Natural Science Foundation of China (grants U1533122 and 51773094), the National Robotics Pro-gramme (Grant 172 25 00063) funded by A*STAR-SERC, Singapore, the Natural Science Foundation of Tianjin (grant 18JCZDJC36800), the Science Foundation for Dis-tinguished Young Scholars of Tianjin (grant 18JCJQJC46600), the Fundamental Research Funds for the Central Universities (grant 63171219), the State Key Laboratory for Modification of Chemical Fibres and Polymer Materials, Donghua University LK1704, the Fundamental Research Funds for the Central Universities (grant 63191139), the National Science Foundation (grant CMMI-1727960).

## Author contributions
Z.F.L., Y.D., and S.F. were responsible for the experimental concept and design. Y.D. and W.H., T.J., K.W., X.Z., X.H., and J.L. carried out the most experiments, characterization and data analyses. Z.W., Z.J.L., E.G., and D.Q. contributed to theoretical simulation and calculation. P.S. carried out the solid-state NMR experiments and data analyses. Z.F.L. was responsible for project administration, conceptualization, supervision, formal ana-lysis, funding acquisition, validation, writing original draft, review and editing. All authors wrote the paper. All authors provided comments and agreed with the final form of the manuscript.

## Competing interests
The authors declare no competing interests.
