## [Peer Review File · Nature Communications]

Reviewers' comments:

Reviewer #1 (Remarks to the Author):

NCOMMS-19-12885

The authors have successfully mimicked spider silk properties (tensile strength, stretchability, toughness, damping capacity etc) by using a simple synthetic polymer and silica nanoparticles DNH including metal ions, twist insertion, and core-sheath morphology controlled by humidity. Each idea is a quite interesting and unique. A significant effort made a quite unique artificial material. This work is really impressive for the reviewer and audience in polymer community as well as material science. From structural views, the reviewer has several fundamental questions. The authors are using ¹H solid state NMR and FT-IR to characterize hydrogen bonding states and ion cross linking. FT-IR would be difficult to quantitatively probe presence and absence of hydrogen bonding and ion-polymer interaction. The reviewer suggests to measure ¹H-¹H DQ NMR spectra to probe the hydrogen bonding structure of PAA. One of the authors did in his past research. Also, the same approach might be useful to prove destructions of hydrogen bonding of PAA when the metal ion was included in the system. The reviewer suggests some additional experiments in structural analysis. Additionally, what is cross-linking structure? Can the authors include a plausible model for the cross-linking in a supporting file?

This work has a significant impact on polymer and materials science community. The reviewer recommend this work to be published in Nature communications after adding some supplementary data.

Reviewer #2 (Remarks to the Author):

This paper by Liu and coworkers presents the development of synthetic fibers with mechanical properties similar to spider silk. The fibers are made based on evaporation of a hydrogel, and ion doping is used along with fiber twisting to optimize the mechanical properties. The optimized system exhibits very high stiffness and stretch, resulting in extremely high energy dissipation. These properties result in materials with excellent damping abilities, applicable when significant shock absorption is required, as demonstrated by the "egg drop" video. I think the findings of this paper are very important and it is really interesting; however, I have a relatively long list of questions and comments regarding the manuscript after reading.

1.) My first issue with the paper is referring to the hydrogel that is used as a "double network hydrogel". This term refers to hydrogel systems with a specific set of properties: the DN gel should consist of two interpenetrating networks, one of which is a sacrificial network and the other which is a stretchable network preventing global rupture. When stretched, the sacrificial network fractures dissipating energy and preventing crack propagation. These systems characteristically exhibit significant hysteresis which is not recoverable upon cycling. The system you utilize here, as introduced in Supplemental Figure 5a, consists of vinyl-functionalised silica nanoparticles, copolymerized in the presence of acrylic acid. There is no "double network" like structure here, or at least in no way that is readily obvious to the reader. Which bonds break to dissipate energy? Why is it reversible? These topics should be discussed. If the energy dissipation is due to reversible fracture of bonds between the acrylic acid and silica nanoparticle, this system seems to have more in common with Suo et al.'s work from their 2012 Nature paper than with double network gels. I think the synthesis process should be talked about more directly in the text; as it is now, it feels like it would be hard to duplicate the experiments the authors performed.

2.) Line 86-87. You state "the as-drawn wet fibre is highly elastic...returning to about half its length if immediately released..." To me, this does not sound highly elastic, and actually pretty significantly viscoelastic. For a material to be described as highly elastic, I would expect there to be little relaxation during a stress-relaxation experiment, and no appreciable creep as a function of (at least relatively short) times. I would consider revising how you initially describe these fibers.

3.) Supplemental figure 7 is a plot of "setting time" vs. fiber diameter. Given that the sheath is higher modulus than the core, thicker fibers take longer to set because a larger rigid sheath is required (makes sense). However, I'm wondering how these values are determined? This is not calculated, correct? But measured experimentally? Please explain how this experiment is

performed.

4.) Around line 106-110. You discuss the buckling surface morphology that is seen when exposed to moisture. I think this is due to the sheath effectively becoming thinner, as the peripheries become more core-like. You say it stems "from the moisture-induced decrease in sheath modulus..." but I disagree somewhat (this is a delicate point, so it's not complete disagreement). It seems that you have two true modulus values, the core (fully swelled hydrogel) and the sheath (fully dried hydrogel). As you expose the fiber to moisture, you transition some of the sheath back into the core, and therefore the STIFFNESS of the sheath decreases, due to a decrease in geometry (rather than a change in modulus).

5.) This brings up a significant comment I have regarding the paper, which is there is no discrete mechanical characterization of the "core" and "sheath". I think that upon knowing the mechanical properties of these two phases, you should be able to develop a simple model to know the required sheath thickness to balance out the force generated from stretching the core. Could you simply cast the hydrogel, and then use some type of micro-indentation or contact mechanics technique to measure the mechanical properties of the gel in the fully-soaked and fully-dried conditions? Knowing the upper and lower bound limits will go a long way in helping understand the results of this materials system.

6.) If you know the modulus of each phase, you can create a simple model to find the required sheath thickness to balance the stiffness of the core as a function of diameter, and compare to the experimentally determined thicknesses, or at least relate to the required drying times.

7.) How does the incorporation of ions result in increased crosslinking? This isn't discussed. In the paper at line 133, you say that metal ions can be used to improve the mechanical properties of hydrogels, but the cited paper refers to polyampholyte hydrogels that have no free metal ions.

8.) If you do transmission electron microscopy, can you see if the heavy metal ions are uniformly dispersed through both the sheath and core? Do they preferentially migrate to one region? Is there any method to visualize the core versus sheath? You mention the sheath is opaque and the core is transparent; can you take a cross-sectional image?

9.) Line 196: you discuss the "fracture toughness" here. The tests you are doing involve measuring the strain energy density, a measure of "toughness". Generally, fracture toughness refers to a materials ability to resist fracture in the presence of a crack or flaw and is characterized by the critical strain energy release rate. A slight difference in terminology, I would probably change this to just "toughness".

10.) The supplemental videos are a nice addition, and clearly demonstrate the "synthetic spider-silk" like properties of the developed materials. In Supplementary Video 4, I was confused about how the test was performed. The net was placed into a chamber at 60% humidity, then the weight was placed, or was humidity applied after the weight was removed? At this point, a time lapse occurred? If so, make sure to list how much the video was increased. It might be better to have a running timer at the bottom, displaying elapsed time.

11.) A few times throughout the paper you refer to the "dragline silk" ...does this refer to the natural silk? Can you just use the terminology "natural spider silk" to prevent confusion?

I'd like to thank the authors for submitting this paper; it's very interesting work and upon revision will be a significant addition to the literature.

Reviewer #3 (Remarks to the Author):

This work reports a hydrogel fibre with spider silk-like properties. The hydrogel fibre was drawn from PAA crosslinked by VSNPs, and further strengthened by metal-ion crosslinking and twist insertion. The resultant PAA hydrogel fibre exhibited very high strength, toughness and impact reduction with low rebound, which is promising in many applications. However, as this fibre is based on very normal PAA matrix, enhanced with conventional crosslinking methods, which have been previously reported by many literatures, I don't think it can be published at this stage unless more comprehensive data and mechanisms are given to support its ultra-high performances. Below are my concerns.

1. What was the exact drying time for the as-prepared fibres? According to experimental section and Fig. S7b, it only required several hundreds of seconds to reach mechanical balance. Whereas, according to Fig. 1b, drying time can be extended to 2h and the core-sheath ratio can still be

substantially affected, was it mechanically unbalanced?

2. Followed by the previous question, if the fibres are dried for longer time, will they dry out? If not, why?

3. How could the core-sheath structure form upon drying at ambient condition? Is it a common phenomenon applicable to other hydrogel materials, or it has something to do with specific drawing technique?

4. Since the hydrogel fibres are extremely sensitive to humidity, and their diameter is just roughly controlled by dipping depth, to ensure accuracy and reproducibility, it is suggested that the authors conduct multiple tests and use error bars for some key data of mechanical properties (for optimized samples), such as modulus, fracture toughness, etc.

5. It is well established how crosslinks and double-network affect mechanical properties of hydrogels in literatures. The authors combined covalent crosslinking and physical crosslinking together with core-sheath structure and twist insertion, which has made the whole system very complex. I think the novelty of this paper lies in the core-sheath structure and twist insertion, while the in-depth mechanisms that relate them to mechanical properties are hardly discussed.

6. Supplementary video 5 and 6 illustrated the use of such hydrogel fibre as escape rope and shock-absorbing net for emergencies, while these demonstrations are kind of misleading. Actually, the hydrogel fibre developed by the authors is unlikely to have both high strength and high stretchability at the same time in practical use. Let alone the small original stretchability (around 40% at 40% RH) that does not accord with the demonstrative video, if the fibre is designed for large applications such as escape rope and shock-absorbing net, larger diameter and number of plies are desired, which would significantly decrease stretchability and energy dissipation capability (as the trend of data shown in Fig. S10, and humidity is not given for these data). The demonstration of buffering an egg falling from 1 meter high already requires a fibre up to 100 plies and a 60% RH, it's not hard to imagine how many more plies are needed for practical applications. In addition, the breaking strain, energy dissipation and damping capacity of multi-ply fibres should also be examined.

Reviewer #1 (Remarks to the Author):

1. The authors have successfully mimicked spider silk properties (tensile strength, stretchability, toughness, damping capacity etc) by using a simple synthetic polymer and silica nanoparticles DNH including metal ions, twist insertion, and core-sheath morphology controlled by humidity. Each idea is a quite interesting and unique. A significant effort made a quite unique artificial material. This work is really impressive for the reviewer and audience in polymer community as well as material science.

Our response: Thank you very much for your valuable comments!

2. From structural views, the reviewer has several fundamental questions. The authors are using 1H solid state NMR and FT-IR to characterize hydrogen bonding states and ion cross linking. FT-IR would be difficult to quantitatively probe presence and absence of hydrogen bonding and ion-polymer interaction. The reviewer suggests to measure 1H - 1H DQ NMR spectra to probe the hydrogen bonding structure of PAA. One of the authors did in his past research. Also, the same approach might be useful to prove destructions of hydrogen bonding of PAA when the metal ion was included in the system. The reviewer suggests some additional experiments in structural analysis. Additionally, what is cross-linking structure? Can the authors include a plausible model for the cross-linking in a supporting file? This work has a significant impact on polymer and materials science community. The reviewer recommend this work to be published in Nature communications after adding some supplementary data.

Our response:

We thank the reviewer for the good suggestion. According to the reviewer's suggestion, we have measured 1H - 1H DQ NMR spectra of samples to probe the hydrogen bonding structure of PAA when the metal ion was included in the system. First, 1H MAS spectra of samples with and without ZnCl_2 under 40 kHz fast MAS were measured as shown in Supplementary Fig. 9e. Proton signals of different types of COOH dimers, free COOH and PAA were observed for sample without ZnCl_2 . With the addition of ZnCl_2 , the peak of COOH dimer obviously decreased, and a new strong peak at about 7.9 ppm was observed. This new peak can be assigned to the protons undergoing chemical exchange between free COOH and water as reported in our previous work.¹

Then we measured 1H DQ NMR spectra of samples without and with ZnCl_2 under 40 kHz MAS as shown in Fig. 2e, f. In Fig. 2e for sample without ZnCl_2 , strong diagonal peak at 13 ppm corresponding to the formation of hydrogen-bonded COOH dimers form, and cross peaks between protons of COOH dimer and aliphatic protons of PAA were clearly observed. In Fig. 2f, with the addition of ZnCl_2 , the diagonal peak at 13 ppm of COOH dimers, and cross peaks between COOH dimer and aliphatic protons (thick red line in Fig. 2e) obviously decreased, indicating the destructions of hydrogen-bonded COOH dimers. With the application of DQ filter in 1H DQ experiment, the strong proton signals under chemical exchange at 7.9 ppm shown in Supplementary Fig. 9e was greatly reduced due to their relatively high mobility, while the less mobile free COOH signal was retained. The predominant cross peaks between protons of free

COOH and aliphatic groups of PAA were observed (thick red line in Fig. 2f). In short, the 2D ^1H DQ NMR spectra clearly prove the destructions of hydrogen bonding of PAA when the metal ion was included in the system.

From the above results, we provide a schematic model for the physical cross-linking of hydrogen bonding of PAA chains before and after addition of ZnCl_2 , as shown in Supplementary Fig. 9f. The above results were added into the revised manuscript and SOM.

References

- Li, B.; Xu, L.; Wu, Q.; Chen, T.; Sun, P.; Jin, Q.; Ding, D.; Wang, X.; Xue, G.; Shi, A. C., Various types of hydrogen bonds, their temperature dependence and water-polymer interaction in hydrated poly(acrylic acid) as revealed by H-1 solid-state NMR spectroscopy. *Macromolecules* **2007**, *40*, 5776-5786.

Supplementary Figure 9. (e) ^1H MAS spectra at 40 kHz MAS of samples without and with ZnCl_2 . (f) Schematic demonstration of physical cross-linking of hydrogen bonding of polyacrylic chains and ionic cross-linking by Zn^{2+} , in addition to the covalent network by vinyl functionalized silica nanoparticles.

Figure 2. (e, f) ^1H double-quantum/single-quantum (DQ/SQ) chemical shift correlation spectra at 40 kHz MAS of samples: (e) without and (f) with ZnCl_2 . Two rotor period of BABA recoupling were used for the excitation and reconversion of DQ coherence.

Reviewer #2 (Remarks to the Author):

This paper by Liu and coworkers presents the development of synthetic fibres with mechanical properties similar to spider silk. The fibres are made based on evaporation of a hydrogel, and ion doping is used along with fibre twisting to optimize the mechanical properties. The optimized system exhibits very high stiffness and stretch, resulting in extremely high energy dissipation. These properties result in materials with excellent damping abilities, applicable when significant shock absorption is required, as demonstrated by the “egg drop” video. I think the findings of this paper are very important and it is really interesting; however, I have a relatively long list of questions and comments regarding the manuscript after reading.

Our response: Thank you very much for this valuable suggestion.

1.) My first issue with the paper is referring to the hydrogel that is used as a “double network hydrogel”. This term refers to hydrogel systems with a specific set of properties: the DN gel should consist of two interpenetrating networks, one of which is a sacrificial network and the other which is a stretchable network preventing global rupture. When stretched, the sacrificial network fractures dissipating energy and preventing crack propagation. These systems characteristically exhibit significant hysteresis which is not recoverable upon cycling. The system you utilize here, as introduced in Supplemental Figure 5a, consists of vinyl-functionalised silica nanoparticles, copolymerized in the presence of acrylic acid. There is no “double network” like structure here, or at least in no way that is readily obvious to the reader. Which bonds break to dissipate energy? Why is it reversible? These topics should be discussed. If the energy dissipation is due to reversible fracture of bonds between the acrylic acid and silica nanoparticle, this system seems to have more in common with Suo et al.’s work from their 2012 Nature paper than with double network gels.

Our response:

We thank the reviewer for the valuable suggestion. Indeed, the polyacrylic hydrogel in this manuscript contains hydrogen-bonding and covalently cross-linked networks, which is similar to the hydrogel work by Prof. Suo et al², in *Nature*, **2012**, 489, 133. The covalent network is formed by cross-linking of vinyl groups in acrylic acid and in vinyl-functionalised silica nanoparticles, which can be deformed under stress and return to the initial state after stress relaxation. During deformation, the hydrogen-bonding cross-links unzip and dissipate energy; after the gel recovers the shape, the hydrogen-bonding re-zips again. By adding ZnCl₂, we observed a dramatic increase in mechanical strength. As proved by our NMR analysis, Zn²⁺ forms ionic cross-links with the COO⁻ groups on the polyacrylic chains. This is another ionic cross-links, which also dissipate energy during mechanical deformation by unzipping and re-zipped after stress-relaxation. We have discussed these in the main text, and we revised the manuscript accordingly.

I think the synthesis process should be talked about more directly in the text; as it is now, it feels like it would be hard to duplicate the experiments the authors performed.

Our response:

We added the synthesis process of the polyacrylic hydrogel fibre into Fig. 1a, b, and we added a brief description of the synthesis process into the text.

2.) Line 86-87. You state “the as-drawn wet fibre is highly elastic...returning to about half its length if immediately released...” To me, this does not sound highly elastic, and actually pretty significantly viscoelastic. For a material to be described as highly elastic, I would expect there to be little relaxation during a stress-relaxation experiment, and no appreciable creep as a function of (at least relatively short) times. I would consider revising how you initially describe these fibres.

Our response:

We agree with the reviewer for this comment. The fibre is actually very viscoelastic. We changed the corresponding description, as follows. “The as-drawn wet fibre is significantly viscoelastic, and a thin fibre (e.g. 20 μm in diameter) will return to about half its length if immediately releasing the stretch.”

3.) Supplemental figure 7 is a plot of “setting time” vs. fibre diameter. Given that the sheath is higher modulus than the core, thicker fibres take longer to set because a larger rigid sheath is required (makes sense). However, I’m wondering how these values are determined? This is not calculated, correct? But measured experimentally? Please explain how this experiment is performed.

Our response:

Yes, the “setting time” was experimentally measured by mounting the as-stretched hydrogel fibre on a homemade holder, and exposing it in ambient air of 40% humidity. The time that the fibre length did not change when releasing one end of the fibre was determined as the “setting time”. We clarified this in the caption of the Supplementary Fig. 7.

4.) Around like 106-110. You discuss the buckling surface morphology that is seen when exposed to moisture. I think this is due to the sheath effectively becoming thinner, as the peripheries become more core-like. You say it stems “from the moisture-induced decrease in sheath modulus...” but I disagree somewhat (this is a delicate point, so it’s not complete disagreement). It seems that you have two true modulus values, the core (fully swelled hydrogel) and the sheath (fully dried hydrogel). As you expose the fibre to moisture, you transition some of the sheath back into the core, and therefore the STIFFNESS of the sheath decreases, due to a decrease in geometry (rather than a change in modulus).

Our response:

We agree with the reviewer for this comment. The fibres studied in this work are exposed in environments under various humidity levels. The sheath is formed from the fibre surface due to water molecule evaporation. From the material point of view, the sheath and the core are formed by the same polymer, with less water content in sheath compared to the core. The “sheath” is identified since its mechanical properties become relatively different with the “core” which is swelled hydrogel. So it is reasonable that on absorbing water molecules, some of the sheath is transitioned back to the core, due to increase in water content. We have changed this in the revised manuscript. We also modified the model to fit the mechanical properties and the geometric dimensions of the hydrogel fibres for different exposure time in ambient air with 40% humidity.

5.) This brings up a significant comment I have regarding the paper, which is there is no discrete mechanical characterization of the “core” and “sheath”. I think that upon knowing the mechanical properties of these two phases, you should be able to develop a simple model to know the required sheath thickness to balance out the force generated from stretching the core. Could you simply cast the hydrogel, and then use some type of micro-indentation or contact mechanics technique to measure the mechanical properties of the gel in the fully-soaked and fully-dried conditions? Knowing the upper and lower bound limits will go a long way in helping understand the results of this materials system.

Our response:

We agree with the reviewer that there is no discrete mechanical characterization between the “core” and the “sheath”. We have addressed this issue in the manuscript by stating that there are transition regions existing between the “sheath” and “core” in hydrogel fibre.

It is a good comment to provide upper bound and lower bound for theoretical modelling the sheath-core structure of the hydrogel fibre. In our experiments, the freshly-drawn hydrogel fibre can be considered a lower bound (fibre core). We have obtained the mechanical properties of the hydrogel fibres exposed in ambient air with 40% relative humidity for different time. Because there is water absorption/desorption balance between the fibre sheath and ambient air, it is reasonable to obtain the same mechanical properties of the fibre sheath that would be used for the upper bound. So the hydrogel fibres were exposed in ambient air with 40% relative humidity for 2 and 4 hours, which almost show the same mechanical properties until the yielding point

(Supplementary Fig. 10f). This indicates that these fibres almost reached the equilibrium state. Metallographic microscope in a reflective mode shows that the hydrogel fibre with drying time of 2 h and 4 h has almost the same fibre sheath content (85%) (Fig. S5c,d). Therefore, the fibre with 2 h drying time was used as the upper bound to theoretically model the mechanical properties of the fibres exposed in ambient air with 40% relative humidity for different time.

Supplementary Fig. S5. (c) Metallographic microscopy images of the hydrogel fibre exposed in ambient air (40% humidity) for different time in a reflective mode. Scale bar: 50 μm .

Supplementary Figure 10. (f) The mechanical properties of hydrogel fibres with different drying time. The black, red and blue curves are for hydrogel fibres with drying time of 0, 2, and 4 h after exposure in ambient air with 40% relative humidity. The hydrogel fibres contained 0.1 wt% VSNPs, without inserted twist or salt, and the original diameter of fibres was 50 μm . The deformation rate was 1.1% s^{-1} .

6.) If you know the modulus of each phase, you can create a simple model to find the required sheath thickness to balance the stiffness of the core as a function of diameter, and compare to the experimentally determined thicknesses, or at least relate to the required drying times.

Using this idea of upper bound and lower bound, we then did a numerical simulation of the mechanical properties for hydrogel fibre at 40% humidity for different exposure time, using the freshly-drawn hydrogel fibre as the lower bound (fibre core), and using the fibre dried at 40% humidity for 2 h as an upper bound (Supplementary Note 5). This allowed us to obtain good agreement of numerical simulation of stress-strain curve with experimental data for hydrogel fibres with different exposure time, as shown in Fig. 2c. The calculated sheath thicknesses for fibres with different drying time are similar to the optically measured values (Supplementary Figure 5d), which increases with drying time. This indicates that the fibre core partially converts to fibre sheath as the drying time increases.

Figure 2. (c) Numerical simulation results of the engineering stress-strain curve of the hydrogel fibres (dotted line) with the same diameter but different wt drying time; the solid line represents the experimental data. The hydrogel fibres contained 0.1 wt% VSNPs, without inserted twist or salt, and the original diameter of fibres was 50 μm . The deformation rate was 1.1% s^{-1} .

Supplementary Figure 5. (d) Optically-measured (inset) and theoretically calculated fiber sheath ratio.

7.) *How does the incorporation of ions result in increased crosslinking? This isn't discussed. In the paper at line 133, you say that metal ions can be used to improve the mechanical properties of hydrogels, but the cited paper refers to polyampholyte hydrogels that have no free metal ions.*

Our response:

We added a new literature² (Suo et al, *Nature*, **2012**, 489, 133) to show incorporation of ions results in increased cross-linking. We also did NMR measurement to show the incorporation of metal ions formed new ionic cross-links. As follows:

We measured ¹H-¹H DQ NMR spectra of samples to probe the hydrogen bonding structure of PAA when the metal ion was included in the system. First, ¹H MAS spectra of samples with and without ZnCl₂ under 40 kHz fast MAS were measured as shown in Supplementary Fig. 9e. Proton signals of different types of COOH dimers, free COOH and PAA were observed for sample without ZnCl₂. With the addition of ZnCl₂, the peak of COOH dimer obviously decreased, and a new strong peak at about 7.9 ppm was observed. This new peak can be assigned to the protons undergoing chemical exchange between free COOH and water as reported in our previous work.¹

Then we measured ¹H DQ NMR spectra of samples without and with ZnCl₂ under 40 kHz MAS as shown in Fig. 2e, f. In Fig. 2e for sample without ZnCl₂, strong diagonal peak at 13 ppm corresponding to the formation of hydrogen-bonded COOH dimers form, and cross peaks between protons of COOH dimer and aliphatic protons of PAA were clearly observed. In Fig. 2f, with the addition of ZnCl₂, the diagonal peak at 13 ppm of COOH dimers, and cross peaks between COOH dimer and aliphatic protons (thick red line in Fig. 2e) obviously decreased, indicating the destructions of hydrogen-bonded COOH dimers. With the application of DQ filter in ¹H DQ experiment, the strong proton signals under chemical exchange at 7.9 ppm shown in Supplementary Fig. 9e was greatly reduced due to their relatively high mobility, while the less mobile free COOH signal was retained. The predominant cross peaks between protons of free COOH and aliphatic groups of PAA were observed (thick red line in Fig. 2f). In short, the 2D ¹H

DQ NMR spectra clearly prove the destructions of hydrogen bonding of PAA when the metal ion was included in the system. From the above results, we provide a schematic model for the physical cross-linking of hydrogen bonding of PAA chains before and after addition of ZnCl_2 , as shown in Supplementary Fig. 9f. The above results were added into the revised manuscript and SOM.

Supplementary Figure 9. (e) ^1H MAS spectra at 40 kHz MAS of samples without and with ZnCl_2 . (f) Schematic demonstration of physical cross-linking of hydrogen bonding of polyacrylic chains and ionic cross-linking by Zn^{2+} , in addition to the covalent network by vinyl functionalized silica nanoparticles.

Figure 2. (e, f) ^1H double-quantum/single-quantum (DQ/SQ) chemical shift correlation spectra at 40 kHz MAS of samples: (e) without and (f) with ZnCl_2 . Two rotor period of BABA recoupling were used for the excitation and reconversion of DQ coherence.

References:

- Li, B.; Xu, L.; Wu, Q.; Chen, T.; Sun, P.; Jin, Q.; Ding, D.; Wang, X.; Xue, G.; Shi, A. C., Various types of hydrogen bonds, their temperature dependence and water-polymer interaction in hydrated poly(acrylic acid) as revealed by H-1 solid-state NMR spectroscopy. *Macromolecules* **2007**, *40*, 5776-5786.
- Sun, J.-Y. *et al.* Highly stretchable and tough hydrogels. *Nature* **489**, 133–136 (2012).

8.) *If you do transmission electron microscopy, can you see if the heavy metal ions are uniformly dispersed through both the sheath and core? Do they preferentially migrate to one region? Is there any method to visualize the core versus sheath? You mention the sheath is opaque and the core is transparent; can you take a cross-sectional image?*

Our response:

We did transmission electron microscopy of the cross-section of hydrogel fibre doped with ZnCl_2 , and elemental mapping shows that the Zn^{2+} ions uniformly distributed in the cross-section of the fibre (Supplementary Fig. S5b).

According to the reviewer's comments, we exposed the hydrogel fibre in ambient air (40% humidity) for different time, and observed these fibres using metallographic microscope in a reflective mode, as shown in Supplementary Fig. S5c. The sheath-core structure is clearly observed, which shows increased thickness of fibre sheath and decreased thickness of fibre core with increasing the exposure time.

Supplementary Figure 5. (b) Elemental mapping shows that Zn^{2+} ions uniformly distributed in the cross-section of the fibre. Scale bar: 2 μm .

Supplementary Fig. S5. (c) Metallographic microscopy images of the hydrogel fibre exposed in ambient air (40% humidity) for different time in a reflective mode. Scale bar: 50 μm .

9.) Line 196: you discuss the “fracture toughness” here. The tests you are doing involve measuring the strain energy density, a measure of “toughness”. Generally, fracture toughness refers to a materials ability to resist fracture in the presence of a crack or flaw and is characterized by the critical strain energy release rate. A slight difference in terminology, I would probably change this to just “toughness”.

Our response:

We thank the reviewer for this kind suggestion. The “fracture toughness” was changed to “toughness” in the revised manuscript.

10.) The supplemental videos are a nice addition, and clearly demonstrate the “synthetic spider-silk” like properties of the developed materials. In Supplementary Video 4, I was confused about how the test was performed. The net was placed into a chamber at 60% humidity, then the weight was placed, or was humidity applied after the weight was removed? At this point, a time lapse occurred? If so, make sure to list how much the video was increased. It might be better to have a running timer at the bottom, displaying elapsed time.

Our response:

The net was placed into a chamber at 60% humidity, then the weight was placed; after the hydrogel net deformed to a stable state, the weight was removed. During this process, the humidity was kept at 60%. A running timer was added into the video to display the elapsed time.

11.) A few times throughout the paper you refer to the “dragline silk” ...does this refer to the natural silk? Can you just use the terminology “natural spider silk” to prevent confusion? I’d like to thank the authors for submitting this paper; it’s very interesting work and upon revision will be a significant addition to the literature.

Our response:

We thank the reviewer for this kind comment. The “dragline silk” was changed to “natural spider silk”.

Reviewer #3 (Remarks to the Author):

This work reports a hydrogel fibre with spider silk-like properties. The hydrogel fibre was drawn from PAA crosslinked by VSNPs, and further strengthened by metal-ion crosslinking and twist insertion. The resultant PAA hydrogel fibre exhibited very high strength, toughness and impact reduction with low rebound, which is promising in many applications. However, as this fibre is based on very normal PAA matrix, enhanced with conventional crosslinking methods, which have been previously reported by many literatures, I don't think it can be published at this stage unless more comprehensive data and mechanisms are given to support its ultra-high performances. Below are my concerns.

Our response:

We thank the reviewer for the valuable comments.

1. What was the exact drying time for the as-prepared fibres? According to experimental section and Fig. S7b, it only required several hundreds of seconds to reach mechanical balance. Whereas, according to Fig. 1b, drying time can be extended to 2h and the core-sheath ratio can still be substantially affected, was it mechanically unbalanced?

Our response:

We thank the reviewer for pointing out this point. To clarify, we changed the terminology “drying time” in Supplementary Fig. 7b to “setting time”. In the figure caption of Supplementary Fig. 7, we defined the “setting time” as the minimum drying time that needed for the fiber to reach the mechanical balance (set the shape after relaxing tethering). The drying time in Fig. 1d in the revised manuscript (Fig. 1b in the previous version) is not the setting time. For drying times of 0.5, 1, and 2h, the fibres are mechanically balanced.

2. Followed by the previous question, if the fibres are dried for longer time, will they dry out? If not, why?

Our response:

We extended the drying time of a 50- μm -diameter hydrogel fibre in ambient air (40% humidity) to 2 h and to 4 h, which almost show the same mechanical properties until the yielding point. This shows that the mechanical properties of the hydrogel fibre almost keeps constant after a certain drying time, and the metallographic microscope in a reflective mode shows that the fibre did not dry out in ambient air (Supplementary Fig. 5c). This indicates that desorption and absorption of water molecules in air into hydrogel fibre reach an equilibrium in ambient air.

Supplementary Figure 5. (c) Metallographic microscopy images of the hydrogel fibre exposed in ambient air (40% humidity) for different time in a reflective mode. Scale bar: 50 μm .

Supplementary Figure 10. (f) The mechanical properties of hydrogel fibres with different drying time. The black, red and blue curves are for hydrogel fibres with drying time of 0h, 2 h, and 4 h after exposure in ambient air with 40% relative humidity. The hydrogel fibres contained 0.1 wt% VSNNPs, without inserted twist or salt, and the original diameter of fibres was 50 μm . The deformation rate was 1.1% s^{-1} .

3. How could the core-sheath structure form upon drying at ambient condition? Is it a common phenomenon applicable to other hydrogel materials, or it has something to do with specific drawing technique?

Our response:

We thank the reviewer for this valuable comment. The fibres studied in this work are exposed in environments under various humidity levels. The sheath is formed from the fibre

surface due to water molecule evaporation. From the material point of view, the sheath and the core are formed by the same polymer, with less water content in sheath compared to the core. The “sheath” is identified since its mechanical properties become relatively different with the “core”, which is swelled hydrogel. So this core-sheath structure is formed spontaneously when the fibre dries in ambient air, and on exposure to high humidity some of the sheath absorbs water molecules and transitions back into the core.

To check if this sheath-core structure also presented in other hydrogel materials or if this drawing technique is necessary for this sheath-core structure, we prepared polyacrylamide/alginate hydrogel according to the literature¹, and prepared a 375- μm -diameter fibre using a polypropylene tube (inner diameter of 0.40 mm, and outer diameter of 0.70 mm) as the template. Core-sheath structures were also observed under metallographic microscopy images in reflective mode. The corresponding stress-strain curves for these fibres show that the breaking strength and modulus of the fibre increases with increasing the drying time. This indicates that the sheath-core structure of the hydrogel fibre can also be observed in other hydrogel materials and without using a drawing technique. We added the above discussions and results in the revised manuscript and supporting information.

Reference:

1. Sun, J.-Y. *et al.* Highly stretchable and tough hydrogels. *Nature* **489**, 133–136 (2012).

Supplementary Figure 18. (a) Metallographic microscopy images in reflective mode of polyacrylamide/alginate hydrogel fibre exposed in ambient air (40% humidity) for different time. Scale bar: 200 μm . The hydrogel is prepared according to literature¹, the fibre was prepared using a polypropylene tube (inner diameter of 0.40 mm, and outer diameter of 0.70 mm) as the template. (b) Stress-strain curves of the polyacrylamide/alginate hydrogel fibres in (a).

4. Since the hydrogel fibres are extremely sensitive to humidity, and their diameter is just roughly controlled by dipping depth, to ensure accuracy and reproducibility, it is suggested that the authors conduct multiple tests and use error bars for some key data of mechanical properties (for optimized samples), such as modulus, fracture toughness, etc.

Our response:

We thank the reviewer for this valuable comment. In the revised manuscript, error bars were added into Fig. 1k, 2b, and 3b, and Supplementary Fig. 1a, 1d, 9b, 10d, 10e, 11b, 12 and 14b to indicate these key values of mechanical strength, strain, modulus, and toughness.

5. It is well established how crosslinks and double-network affect mechanical properties of hydrogels in literatures. The authors combined covalent crosslinking and physical crosslinking together with core-sheath structure and twist insertion, which has made the whole system very complex. I think the novelty of this paper lies in the core-sheath structure and twist insertion, while the in-depth mechanisms that relate them to mechanical properties are hardly discussed.

Our response:

We thank the reviewer for this valuable comment. We added more experiments and theoretical modelling to understand the mechanisms related to mechanical properties of the hydrogel fibre, including (1) enhanced ionic cross-linking by adding Zn^{2+} by using solid state NMR; (2) refined model of core-sheath structure by using measured upper bound (dry fibre) and lower bound (wet fibre); (3) microscopic characterization and modelling of mechanical properties hydrogel fibre by the twist insertion. As follows:

1. NMR characterization of formation of ionic cross-link by adding Zn^{2+}

First, we measured ^1H - ^1H DQ NMR spectra of samples to probe the hydrogen bonding structure of PAA when the metal ion was included in the system. First, ^1H MAS spectra of samples with and without ZnCl_2 under 40 kHz fast MAS were measured as shown in Supplementary Fig. 9e. Proton signals of different types of COOH dimers, free COOH and PAA were observed for sample without ZnCl_2 . With the addition of ZnCl_2 , the peak of COOH dimer obviously decreased, and a new strong peak at about 7.9 ppm was observed. This new peak can be assigned to the protons undergoing chemical exchange between free COOH and water as reported in our previous work.²

Then we measured ^1H DQ NMR spectra of samples without and with ZnCl_2 under 40 kHz MAS as shown in Fig. 2e, f. In Fig. 2e for sample without ZnCl_2 , strong diagonal peak at 13 ppm corresponding to the formation of hydrogen-bonded COOH dimers form, and cross peaks between protons of COOH dimer and aliphatic protons of PAA were clearly observed. In Fig. 2f, with the addition of ZnCl_2 , the diagonal peak at 13 ppm of COOH dimers, and cross peaks between COOH dimer and aliphatic protons (thick red line in Fig. 2e) obviously decreased, indicating the destructions of hydrogen-bonded COOH dimers. With the application of DQ filter in ^1H DQ experiment, the strong proton signals under chemical exchange at 7.9 ppm shown in Supplementary Fig. 9e was greatly reduced due to their relatively high mobility, while the less mobile free COOH signal was retained. The predominant cross peaks between protons of free

COOH and aliphatic groups of PAA were observed (thick red line in Fig. 2f). In short, the 2D ^1H DQ NMR spectra clearly prove the destructions of hydrogen bonding of PAA when the metal ion was included in the system. From the above results, we provide a schematic model for the physical cross-linking of hydrogen bonding of PAA chains before and after addition of ZnCl_2 , as shown in Supplementary Fig. 9f.

Reference:

2. Li, B.; Xu, L.; Wu, Q.; Chen, T.; Sun, P.; Jin, Q.; Ding, D.; Wang, X.; Xue, G.; Shi, A. C., Various types of hydrogen bonds, their temperature dependence and water-polymer interaction in hydrated poly(acrylic acid) as revealed by H-1 solid-state NMR spectroscopy. *Macromolecules* **2007**, *40*, 5776-5786.

Supplementary Figure 9. (e) ^1H MAS spectra at 40 kHz MAS of samples without and with ZnCl_2 . (f) Schematic demonstration of physical cross-linking of hydrogen bonding of polyacrylic chains and ionic cross-linking by Zn^{2+} , in addition to the covalent network by vinyl functionalized silica nanoparticles.

Figure 2. (e, f) ^1H double-quantum/single-quantum (DQ/SQ) chemical shift correlation spectra at 40 kHz MAS of samples: (e) without and (f) with ZnCl_2 . Two rotor period of BABA recoupling were used for the excitation and reconversion of DQ coherence.

(2) Numerical simulation of core-sheath fibre at different drying time

We did a numerical simulation of the mechanical properties for hydrogel fibre at 40% humidity for different exposure time, using the freshly-drawn hydrogel fibre with drying time of 0 h as the lower bound (fibre core), and using the fibre dried at 40% humidity for 2 h as an upper bound (Supplementary Note 5). This allowed us to obtain good agreement of numerical simulation of stress-strain curve with experimental data for hydrogel fibres with different exposure time, as shown in Fig. 2c. The calculated sheath thicknesses for fibres with different drying time are similar to the optically measured values, which increases with drying time. This indicates that the fibre core partially converts to fibre sheath as the drying time increased.

Figure 2. (c) Numerical simulation results of the engineering stress-strain curve of the hydrogel fibres (dotted line) with the same diameter but different drying time; the solid line represents the experimental data. The hydrogel fibres contained 0.1 wt% VSNPs, without inserted twist or salt, and the original diameter of fibres was 50 μm . The deformation rate was 1.1% s^{-1} .

Supplementary Figure 5. (d) Optically-measured (inset) and theoretically calculated fiber sheath ratio.

(3) Microscopic characterization and modelling of mechanical properties hydrogel fibre by the twist insertion.

We first investigated the morphology change of hydrogel fibres during twist insertion, as shown in Supplementary Fig. S4a. It can be seen that with twist insertion, the hydrogel fibres get flattened, and the inner core get less transparent. This indicates that the fibre gets more sheath-like, possibly due to the decreased thickness in the flattened region. This corresponds to the increased mechanical stretch and modules of the hydrogel fibre with increasing of inserted twist. Finite elemental modelling (FEM) simulations were carried out to validate the effect of twist insertion on the internal residual stress/strain in a core-sheath structured hydrogel fibre. A 20- μm -diameter hydrogel fibre (sheath thickness of 3 μm , and core radius of 7 μm) was constructed in FEM. It is found that upon twist insertion (6 turns/mm), periodic internal stress along the fibre length was generated in both fibre sheath and fibre core, resulting in flattening of the hydrogel fibre (Supplementary Fig. 4a). This agrees with the experimental observation of twist-induced flattening of hydrogen fibres in Supplementary Fig. 4c. The FEM also shows that the twisted core-sheath hydrogel fibre showed higher tensile strength and modules than the non-twisted hydrogel fibre (Supplementary Fig. 4b).

Moreover, the length of polymer chains in the fibre actually elongated by forming spiral structure during twist insertion, this may result in the following two consequences in the macromolecular level. (1) The random coiled polymer chains were stretched and get aligned in some extent in the twisting direction. Such alignment of polymer chains would result in increased anisotropy of the fibre, increasing the mechanical strength during stretching. (2) Twist-induced elongation of the polymer chain may generate internal stress and increase the bond angle in the polyacrylic acid chain, therefore induced increased rigidity of the polymer chains. These twist-induced change in the macromolecular level may contribute to increase in mechanical strength.

Supplementary Figure 4. (a) Metallographic microscopy images of the hydrogel fibres with different inserted twist. Scale bar: 40 μm . (b) FEM modelling showing stress distribution on the sheath of a 20- μm -diameter core-sheath hydrogel fibre. The inserted twist is 6 turns/mm, the sheath thickness is 3 μm , and core radius is 7 μm in FEM modelling.

6. *Supplementary video 5 and 6 illustrated the use of such hydrogel fibre as escape rope and shock-absorbing net for emergencies, while these demonstrations are kind of misleading. Actually, the hydrogel fibre developed by the authors is unlikely to have both high strength and high stretchability at the same time in practical use. Let alone the small original stretchability (around 40% at 40% RH) that does not accord with the demonstrative video, if the fibre is designed for large applications such as escape rope and shock-absorbing net, larger diameter and number of piles are desired, which would significantly decrease stretchability and energy dissipation capability (as the trend of data shown in Fig. S10, and humidity is not given for these data). The demonstration of buffering an egg falling from 1 meter high already requires a fibre up to 100 plies and a 60% RH, it's not hard to imagine how many more plies are needed for practical applications. In addition, the breaking strain, energy dissipation and damping capacity of multi-ply fibres should also be examined.*

Our response:

We agree with the reviewer for the comments. Supplementary video 5 and 6 were removed from manuscript. The relative humidity during mechanical test in Supplementary Fig. 10 is 40%, which was added in supporting information. The breaking strain, energy dissipation and damping capacity of multi-ply fibres were examined and added into the supplementary Table S4, and the breaking stress and breaking strain as a function of number of plies were shown in Supplementary Fig. 10e.

Supplementary Fig. 10. (e) Breaking stress and breaking strain for different number of plies of hydrogel fibres. The deformation rate was $27.8\% \text{ s}^{-1}$ and the twist density was 3 turns mm^{-1} . The relative humidity was 40%.

Supplementary Table 4. Mechanical properties of hydrogel fibres presenting 3 turns mm^{-1} twists at different relative humidity and deformation rates

Hydrogel fibre sample (PAA/VSNPs _y /X _z)	Condition	Breaking stress (MPa)	Breaking strain (%)	Toughness (MJ m ⁻³)	Energy dissipation (MJ m ⁻³)
PAA/VSNPs _{0.1} /ZnCl _{2-0.02}	0.6% s ⁻¹ (^a)	271	67.4	80.5	76.3
PAA/VSNPs _{0.1} /ZnCl _{2-0.02}	1.1% s ⁻¹ (^a)	332	67.3	108	102
PAA/VSNPs _{0.1} /ZnCl _{2-0.02}	2.8% s ⁻¹ (^a)	398	67.2	130	123
PAA/VSNPs _{0.1} /ZnCl _{2-0.02}	4.4% s ⁻¹ (^a)	462	67.3	151	143
PAA/VSNPs _{0.1} /ZnCl _{2-0.02}	5.6% s ⁻¹ (^a)	527	67.3	183	174
PAA/VSNPs _{0.1} /ZnCl _{2-0.02}	11.1% s ⁻¹ (^a)	589	67.4	217	205
PAA/VSNPs _{0.1} /ZnCl _{2-0.02}	27.8% s ⁻¹ (^a)	647	67.4	259	245
PAA/VSNPs _{0.1} /ZnCl _{2-0.02}	10-ply(^b)	614	66.5	241	226
PAA/VSNPs _{0.1} /ZnCl _{2-0.02}	30-ply(^b)	593	56	219	208
PAA/VSNPs _{0.1} /ZnCl _{2-0.02}	50-ply(^b)	554	46	196	182
PAA/VSNPs _{0.1} /ZnCl _{2-0.02}	100-ply(^b)	451	38	147	136
PAA/VSNPs _{0.1} /ZnCl _{2-0.02}	30% RH(^c)	648	67.3	256	243
PAA/VSNPs _{0.1} /ZnCl _{2-0.02}	40% RH(^c)	647	67.4	258	245
PAA/VSNPs _{0.1} /ZnCl _{2-0.02}	50% RH(^c)	512	81.3	235	217
PAA/VSNPs _{0.1} /ZnCl _{2-0.02}	60% RH(^c)	453	117	249	222
PAA/VSNPs _{0.1} /ZnCl _{2-0.02}	70% RH(^c)	399	137	215	172
PAA/VSNPs _{0.1} /ZnCl _{2-0.02}	80% RH(^c)	315	160	155	110
PAA/VSNPs _{0.1} /ZnCl _{2-0.02}	100%RH(^c)	194	207	153	79.6

For (a), the RH was 40% and the twist density was 3 turns mm^{-1} ; for (b), the deformation rate was 27.8% s⁻¹, RH was 40%, and the twist density was 3 turns mm^{-1} ; for (c), the deformation rate was 27.8% s⁻¹, and the twist density was 3 turns mm^{-1} .

REVIEWERS' COMMENTS:

Reviewer #1 (Remarks to the Author):

I recommend this paper to be accepted in Nature communication.

Reviewer #2 (Remarks to the Author):

I'm pleased with the revisions made by the authors. I think that this is a fine paper and very interesting work. I'm happy to recommend it for publication. I just noted a few small points regarding the grammar that stood out and think that a formal proofreading could improve the readability of the final draft.

Line 83: "First" is spelled incorrectly.

Line 110: "As the hydrogel fibre are exposed in environments in ambient air (40% humidity)." Is not a complete sentence and should be combined with the next sentence.

Line 269: "Moreover, the length of polymer chains in the fibre actually elongated by forming spiral structure during twist insertion, this may result in the following two consequences in the macromolecular level." This is a run-on sentence and should be split into two sentences.

These are just some examples, and other improvements should be made, but the science of the paper is sound and I think it will have significant impact in the literature.

Reviewer #3 (Remarks to the Author):

I think my concerns were well addressed and now it can be published.

Point-to-Point Reply to the Reviewers' Comments

Manuscript no. NCOMMS-19-12885A

Title: Ion-Doped and Twisted Core-Sheath Hydrogel Fibres with Spider Silk-Like Strength, Toughness, and Stretchability

Reviewer #1 (Remarks to the Author):

I recommend this paper to be accepted in Nature communication.

Our response: Thank you very much for your kind effort for improving this article.

Reviewer #2 (Remarks to the Author):

I'm pleased with the revisions made by the authors. I think that this is a fine paper and very interesting work. I'm happy to recommend it for publication. I just noted a few small points regarding the grammar that stood out and think that a formal proofreading could improve the readability of the final draft.

Our response: Thank you very much for your kind effort for improving this article.

Line 83: "First" is spelled incorrectly.

Our response: Corrected.

Line 110: "As the hydrogel fibre are exposed in environments in ambient air (40% humidity)." Is not a complete sentence and should be combined with the next sentence.

Our response: This sentence was combined with the next sentence, as follows:

"As the hydrogel fibre are exposed in environments in ambient air (40% humidity), the sheath is formed from the fibre surface due to water molecule evaporation."

Line 269: "Moreover, the length of polymer chains in the fibre actually elongated by forming spiral structure during twist insertion, this may result in the following two consequences in the macromolecular level." This is a run-on sentence and should be split into two sentences.

Our response: This sentence was split into two sentences, as follows:

"Moreover, the length of polymer chains in the fibre actually elongated by forming spiral structure during twist insertion. This may result in the following two consequences in the macromolecular level."

These are just some examples, and other improvements should be made, but the science of the paper is sound and I think it will have significant impact in the literature.

Our response: We have made a fully check of the grammar using “Nature Research Editing Service”.

Reviewer #3 (Remarks to the Author):

I think my concerns were well addressed and now it can be published.

Our response: Thank you very much for your kind effort for improving this article.